# FedGraph: A New Paradigm for Federated Graph Learning

## Abstract

Federated learning is a distributed approach to training a global model over multiple clients without sharing their local data. In graph data, the data heterogeneity can correspond to subgraph structures and node features varying extremely different, and the *model isolation to federated graph task* corresponds to exclusive schema on handing data for specific task in clients, e.g., anomaly user setting in Twitter is rather different from LinkedIn. Although most federated graph learning approaches are employed to address the data heterogeneity challenge, we find that the *model isolation to federated graph task* challenge has been overlooked. This *model isolation to federated graph task* will prevent existing models into the federated graph learning framework. In this paper, we propose FedGraph: a new paradigm for federated graph learning. The key idea is to utilize the graph structure without private node features as *structure knowledge* bridging all task specific knowledge in clients. Our extensive experiments show that FedGraph significantly outperforms the other state-of-the-art federated learning algorithms on anomaly detection tasks. Two deep learning models and one existing anomaly subgraph detection model are transferred to FedGraph framework.

## 1 Introduction

Graph applications are becoming increasingly prevalent in social activities, such as account recommendation, hot event detection, and short video ranking on Twitter, LinkedIn, and TikTok. Most applications developed by these companies deliver personalized content to users based on users' personalized interests inferred from private data. In one social network service (e.g., Twitter), when users exhibit similar behaviors, such as focusing on "US election" tweets, they tend to fall into a similar interest distribution. The extra data from Weibo also fit the personal interest distribution in detail. Moreover, the EU General Data Protection Regulation (GDPR) regulates the collection, storage, and processing of personal data, and has an effect on most major companies worldwide Gruschka et al. (2018).

However, an important challenge raised from graph data heterogeneity is tackled by recent Federated Graph Learning (FGL) methods Baek et al. (2023) from the personalized parameter perspective. In graph data, the number of users features exhibit extremely non-independent and identically (non-IID) properties, and the graph structure also exhibits more heterogeneity. In the federated learning (FL) setting, most Graph Neural Networks (GNNs) consider the structure embedding, and the structure heterogeneity is tackled by the personalized weighted parameters of the deep models Baek et al. (2023); Yang et al. (2021). However, as the size of the graph increases, GNN-based models incur significant communication costs. Additionally, traditional non-GNN models cannot be directly used for federated learning.

Moreover, a significant challenge, *model isolation to federated graph task*, that has been overlooked by the existing FL methods since graph learning models can be easily adapted to different graph structures. The isolation of graph data and computation-specific algorithms has resulted in a gap in performing federated tasks across multiple graph data, referred to as *model isolation to federated graph task*. In Figure 1, data usually exist in the form of isolated islands Yang et al. (2019) in most industries, and the task of detecting trending events on platforms such as Twitter and Weibo is straightforward, but the specific knowledge of hot events (significant structure of the community and users' features) is often achieved within black boxes to protect algorithm execution processes.

Figure 1: An example of a *model isolation to federated graph task challenge*. In hotspot event group mining on Twitter and Weibo, the "US election" is an extremely popular topic on Twitter. However, on Weibo, the top trending topic is the "stock market". Due to data isolation and the privacy of their algorithms, traditional federated learning is limited in performing the federated task.

In traditional federated graph learning, multiple tasks cannot be performed collaboratively if graph data and algorithm are both private. Graph data can be categorized into structures and features. We protect graph features and algorithm execution processes, but the graph structure can be shared. So federated graph learning tasks are aligned on the graph structures.

Motivated by these challenges, we propose a new paradigm for federated graph learning that treats the graph structure as a bridge to private node features (in this paradigm, 'knowledge' refers to subgraphs mined by algorithms). For instance, a data owner (e.g., Twitter) obtains a graph that incorporates its node features, allowing the use of exclusive graph learning methods within the organization. We take the global graph structure as structural knowledge, which can communicate with other data owners. We protect node feature privacy across all data fields. In this new paradigm, we consider node features as privacy, and the graph structure in the server can be accessed by clients. For example, a client may recommend a set of users as influencers on Twitter; however, those users may remain unrecognized by other clients due to the absence of their private features. This paper takes anomaly detection tasks as an example. Unless otherwise specified, anomaly detection will be considered as the downstream task in the following text. For anomaly detection on multiple graphs, limited exploration has been done for federated anomaly detection without data leakage (Zhao et al., 2019; Chen et al., 2019; Ying et al., 2021; Zhao et al., 2020). A well-known cost function $F_\alpha(S)$ can be specifically modeled in the $i$th industry data to anomaly detection task. We employ an alignment function $Q_\sigma(S, U)$ for measuring the distance between subgraphs of $S$ and $U$. $F$ is the abnormal score of $S$ (e.g., the work Tree-Shape-Priors Subgraph Detection (TSPSD) (Wu et al., 2018) as $F$). $Q$ is the alignment score of $S$ and $U$ (e.g., the work Cross-Network Embedding for Multi-Network Alignment (CrossMNA) (Chu et al., 2019) as $Q$). The parameters $\alpha$ and $\sigma$ are significant level and alignment threshold. We refer to this new paradigm as Federated Graph learning (FedGraph).

We summarize our main contributions as follows:

- **Innovative work.** To the best of our knowledge, we are first to study federated graph learning problem without aggregation on parameters, and with performing exclusively algorithms on clients, referred to as *model isolation to federated graph task* challenge. We novel employ the graph structure bridging different models from clients.

- **Broad application.** A broad spectrum of graph learning models can be easily transferred to FEDGRAPH framework. Specifically, for the graph scan statistics (e.g., Berk-Jones) and two GNN models (e.g., Dominant, CoLa), the models are successfully transferred to FEDGRAPH framework in anomaly detection tasks.

- **Effectiveness and robustness.** Extensive experiments on five real datasets have verified that anomaly tasks in FEDGRAPH can be effectively applied to different mulitiple graph settings. Our proposed FEDGRAPH achieved at least $56\%$ accuracy under the $30\%$ noise level.

## 2 RELATED WORK.

**Federated learning.** Federated learning, being an efficient technique for privacy protection and suitable for cross-data, cross-domain, and cross-enterprise applications, is usually categorized into horizontally federated learning, vertically federated learning, and federated transfer learningYang et al. (2019). Kesici et al. (2024) detect real-time false data injection attacks in distribution networks by utilizing split learning-based vertical FL. Qiu et al. (2024) detect the heart sound abnormality in multi-center institutional heart sound databases by aligning the feature spaces of horizontal FL participating institutions. Quan et al. (2024) integrate federated transfer learning and differential privacy to classify diffuse waste data while ensuring data privacy.

**Federated Graph Learning.** In this article, the Federated Graph Learning (FGL) we study refers to standard FL with structured data (i.e., server and clients work on graph data). Existing FGL methods focus on reconstructing cross-client information, embedding overlapping nodes during collaborative training, and dealing with data heterogeneity from multiple clientsFu et al. (2022). For instance, Xie et al. (2021) and He et al. (2021) focus on addressing the challenge of the heterogeneity among non-IID graphs, they assume that different clients have completely disjoint graphs. Li et al. (2020) and Xie et al. (2021) tackle statistical heterogeneity by considering local partial data or server-side proxy data as model parameters. When the graphs owned by clients are parts of an original global graph, Wu et al. (2021) and Zhang et al. (2021) address the missing links and overlapping nodes between subgraphs by augmenting nodes with information from other subgraphs. McMahan et al. (2017) train the decentralized data on mobile devices and learn the shared model by aggregating and averaging locally computed updates. Baek et al. (2023) focus on the joint improvement of local models working on interrelated subgraphs, offering a novel solution to the challenges in subgraph-level FL. However, these approaches can compromise privacy and increase communication overhead. Hence, no published research studies task-specified structured knowledge federated learning. Our method tackles the above challenge differently by exploring subgraph communities.

## 3 PROBLEM FORMULATION

We define our novel problem of federated graph learning with the same graph structure. The problem has been extensively discussed in clients owning various graph-structured data.

**Definition (Model isolation to federated graph task).** A graph $\mathcal{G} = (V, E)$ with $n$ nodes has node feature matrix $\mathbf{X} \in \mathbb{R}^{n \times d}$, where $V = \{v_1, \cdots, v_n\}$ represents vertices and $E \subseteq V \times V$ represents edges. Each vertex $v$ has a $d$-dimensional feature $\mathbf{X}(v, :)$. A local graph $\mathcal{G}_i$ on the $i$-th client can be a global graph on the server: $\mathcal{G}_i = \mathcal{G}$. The global graph on the server is $\mathcal{G}_0$, with each local graph $\mathcal{G}_i$ having a private node feature matrix $\mathbf{X}_i$. Graph learning problem can be transformed into structure (public) learning combined with feature (private) learning in graph Ding et al. (2019). In graph settings $\mathcal{G}_0, \{\mathcal{G}_1, \mathcal{G}_2, \ldots, \mathcal{G}_N\}, \{\mathbf{X}_1, \mathbf{X}_2, \ldots, \mathbf{X}_N\}$, our objective is as follows:

$$\min \sum_{G_i \subseteq \mathcal{G}} \mathcal{L}_i(G_i; \overline{\boldsymbol{S}}), \quad \overline{S}^{(k+1)} = \sum_{i=1}^{N} \overline{S}^{(k)} \oplus S_i \qquad (1)$$

where $\overline{S}$ is the global *structural knowledge* (i.e., the subgraph) in the server, $\oplus$ is the Operators for sets (i.e., $\cap$).$S_i$ is the $i$-th client's structural knowledge. $\mathcal{L}_i \neq \mathcal{L}_j$ indicates that there is model isolation between different clients. In federated graph learning, we observe the two characteristics of model isolation to federated graph task:

**Characteristic 1. (Exclusive Algorithm)** In specific real-world federated task scenarios, client $i$ possesses private data $\mathcal{G}_i$ and a corresponding private algorithm $f_i$. Due to privacy and property rights limitations, clients do not share their full algorithms with one another. For instance, in the task of identifying influential groups on Twitter and LinkedIn, Twitter emphasizes influencers with active audiences, while LinkedIn prioritizes those who graduated from Ivy League universities and work at large companies. The data and mining algorithms used by these platforms are inconsistent, hindering the application of traditional federated learning.

**Characteristic 2. (Parameter Mismatch)** Due to the aforementioned exclusive Privacy scenario, the parameter dimensions and parameter spaces of different algorithm models do not match. For

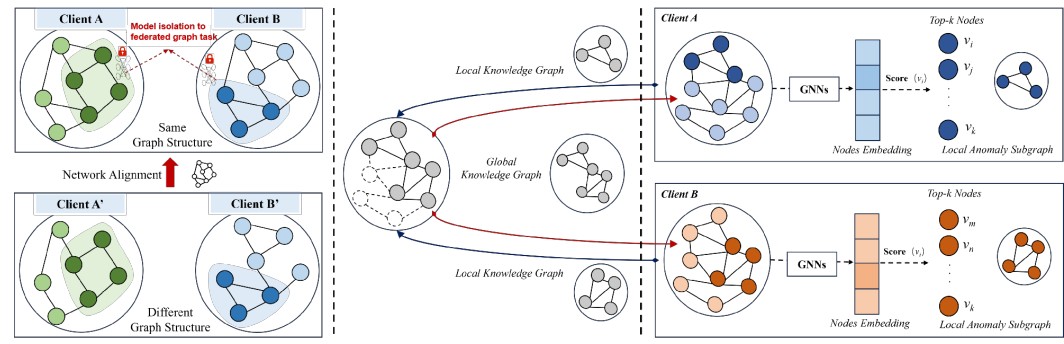

(a) Federated Graph Learning Problem    (b) Adaptive structure knowledge sample in server    (c) Local structure knowledge update in client

Figure 2: An illustration of our proposed overall architecture. (a) The task knowledge of existing graph federated learning work ' Baek et al. (2023); Tan et al. (2023) is isolated from outside. Its models are no space may be left to tune personalized parameters. (b) A new paradigm for federated graph learning. The graph structure is distributed to clients, and local knowledge graphs are aggregated on the server. Network alignment approaches are used to align local graph structures with the global graph in different settings. (c) Each client still runs on its model. These models can be easily transferred to FedGraph framework.

example, for a classification task, a graph owner employs SVM algorithm, and another graph owner employs naive Bayes algorithm, however, there are extremely different parameters from dimensions, rather than aggregate this parameters. Therefore, traditional federated learning methods that rely on parameter sharing cannot effectively address this challenge.

Since the challenge of model isolation to federated graph task, different clients fail to upload their parameters for model aggregation. To address the challenge of model isolation to federated graph task, we propose the new federated learning paradigm in this paper. We focuses on the structure of graph data, using the *structural knowledge* to bridge the knowledge gap between different clients' private data and private algorithms instead of parameters.

The FedGraph problem (1) considers two cases: $\forall i, \mathcal{G}_i = \mathcal{G}_0$ and $\exists i, \mathcal{G}_i \neq \mathcal{G}_0$. For $\mathcal{G}_i = \mathcal{G}_0$, we discuss client GNN models. For $\mathcal{G}_i \neq \mathcal{G}_0$, we explore network alignment between $\mathcal{G}_i$ and $\mathcal{G}_0$, aligning *structure knowledge* $S$ with $S'$ (a subgraph of $\mathcal{G}_i$). Minimizing problem (1) transfers *structure knowledge* from local to global models, e.g., from local anomalies $S' = \{S_i\}$ in clients $\{\mathcal{G}_i, \mathbf{X}_i\}$ to the global anomaly $S$.

## 4 METHODOLOGY

To address the challenge of model isolation to federated graph task, we proposed to perform the proximal local structure knowledge update, and to mask unrelated structure knowledge.

### 4.1 LOCAL STRUCTURE KNOWLEDGE UPDATE WITH PROXIMAL TERM

We aim to capture the local structure knowledge sequence $S_i^1, S_i^2, \cdots$ for each client $i \in [N]$, and the private knowledge $S_i$ is transformed to public knowledge $U_i$ over the global graph $\mathcal{G}_0$ in the server. The local private knowledge $S_i$ is achieved by accessing private feature data $\mathbf{X}_i$ for the $i$-th client, and the public knowledge $U_i$ is achieved by minimizing a distance function $-\lambda Q_\sigma^i(S_i, U)$ where the value of $Q$ is proportional to the number of anchor links between $S_i$ and $U$. The public knowledge is acquired from only the graph structure without accessing the local private node feature data.

#### 4.1.1 PRIVATE LOCAL KNOWLEDGE ESTIMATION

Each client employs an objective function $\mathcal{L}^i$ to detect local structure knowledge by consolidating their private node feature data. The private knowledge at steps c1 and c2 is detected exclusively in the data owner setting. Each client provides meaningful privacy guarantees.

$$S_i^* \leftarrow \arg \min_{S \subseteq \mathcal{G}_i} \mathcal{L}^i(S) + (-\lambda)Q_\sigma^i(S, U) \tag{2}$$

where in the same graph-structure setting, the function $\mathcal{L}^i$ can be reformulated as the loss function (5), and the GNN-based models can be used to identify private local knowledge. The proximal term $(-\lambda)Q_\sigma(S, U)$, (i.e., a distance function between $S$ and $U$), prevents the local knowledge $S_i$ from extremely drifting to the local data distribution. The $Q$ function value is proportional to $|S \cap U|$ since $\forall i, \mathcal{G}_i = \mathcal{G}_0$. The local knowledge $S_i$ is closer to the global knowledge $U$, and the hyperparameter $\lambda$ scales the impacts on the closeness (e.g., $\lambda = 1$). However, in the different same graph-structure setting, the function $\mathcal{L}^i$ is reformulated as the non-parametric graph scanning statistic $-F$ as a loss function, $F_\alpha^i(S) = \varphi(\alpha, N_\alpha(S), N(S))$ where $S$ is a subgraph, and $\alpha$ is the significant level, $N_\alpha(S)$ is the number of anomaly vertices in $S$ whose p-value is less than or equal to $\alpha$, and $N(S)$ is the total number of vertices in $S$. We consider two functions for $F_\alpha^i(S)$: Berk-Jones (BJ) statistic (Berk & Jones, 1979) and Higher Criticism (HC) statistic (Donoho et al., 2004). The proximal term, $-\lambda Q_\sigma^i(S, U)$ can not directly compute the distance between $S$ and $U$ since $\exists i, \mathcal{G}_i \neq \mathcal{G}_0$, and the network alignment based models are required to solve this problem. Then the local private knowledge detection problem can be reformulated as the federated anomaly detection problem (6) over multiple clients with different graph structures.

### 4.1.2 PUBLIC GLOBAL KNOWLEDGE ALIGNMENT

We aim to measure private knowledge $S$ and public knowledge $U$ with network alignment-based models. We define the function $Q$ as follows:

$$Q_\sigma(S, U) = \frac{N_\sigma(S, U)}{N(S)} + \frac{N_\sigma(S, U)}{N(U)} \tag{3}$$

where $\sigma$ is the predefined alignment threshold. The private knowledge $S$ is aligned with the public knowledge subgraph $U$. The two subgraphs are connected. $N_\sigma(S, U)$ is the number of anchor links between $S$ and $U$ whose alignment probability is greater than or equal to $\sigma$, and especially for the same graph structure $N_\sigma(S, U) = |S \cap U|$. $N(S)$ and $N(U)$ are the number of all nodes in $S$ and $U$ respectively. The node's alignment probability is obtained through the network alignment work CrossMNA (Chu et al., 2019). By introducing this algorithm, we pre-aligned each private network with the public network and obtained the alignment probability of all pairs of nodes among them. We use network alignment to map the similarity between the subgraphs to a value and obtain the most similar part between the two subgraphs during this process.

### 4.2 ADAPTIVE STRUCTURE KNOWLEDGE SAMPLING

Based on the previous structure knowledge update scheme, we collect the public knowledge $\{U_i^*\}$ from all clients. Despite transforming client models into graph-structure operations, heterogeneity in federated graph learning persists due to structural differences. Thus, joining all public knowledge $\cup_{i=1}^N U_i^*$ may be inadequate, as it focuses on the relevance of knowledge rather than the relevance of *structure knowledge*. We propose selecting the most relevant public structure knowledge from the aggregated structure knowledge transmitted by clients, akin to existing skills Wang & Zhang (2022).

**Relevant public structure knowledge sampling.** We create a coalition partition $\Pi = \{C_i\}_{i=1}^M$ over the set of $\{U_i^*\}_{i=1}^N$ by computing graph connected components in the global graph $\mathcal{G}_0$. Each $C_i$ is a connected subgraph, and the relevant knowledge $\{U_j^*\}$ are connected into a coalition. We use $\Gamma \in \mathbb{R}^{M \times 1}$ for modeling contributions of each coalition. Notably, For the coalition $C_i$, we compute function $\Gamma(i) = \sum_{U_j^* \in C_i} Q_\sigma^j(S_j^*, U_j^*)$ where just only values of function $Q$ are used to $\Gamma$ that do not violate data privacy constraints. In the same graph structure setting, the value of $Q$ is just proportional to the structure knowledge $U$ where $U$ will be uploaded to the server. The contribution $\Gamma$ is normalized for $\sum_i \Gamma(i) = 1$, and the normalized $\Gamma$ acts as a multinomial distribution. The global structure knowledge is updated as follows:

$$U \leftarrow C_i \qquad \text{for random sampling } i \text{ from } \Gamma \tag{4}$$

The higher probability in $\Gamma$ means the more contribution of the coalition. For each client, the objective function (2) shows that the structure knowledge $U$ is parameterized within the function $Q$. We select the most contribution $U$ as the public structure knowledge.

### 4.3 FEDERATED GRAPH LEARNING ALGORITHM

We propose a new federated learning paradigm, FEDGRAPH, which utilizes structural knowledge sharing instead of traditional parameter sharing. This effectively addresses the two characteristics of model isolation to federated graph task. The main computational steps and process of the FEDGRAPH are described. For the $i$-th participant side (client), given the global structure knowledge $U$ (the subgraph of $\mathcal{G}_0$), we perform the local structure knowledge update as follows:

(c1) $S_i^* \leftarrow \arg\min_{S \subseteq \mathcal{G}_i} \mathcal{L}^i(S) + (-\lambda)Q_\sigma^i(S, U)$;

(c2) $U_i^* \leftarrow \arg\min_{U \subseteq \mathcal{G}_0} -\lambda Q_\sigma^i(S_i^*, U)$;

For the server side, we aggregate the local knowledge $\{U_i^*\}$ for $i \in [N]$,

(s1) Create a coalition partition $\Pi$ over the set of $\{U_i^*\}_{i=1}^N$, where the partition approach is illustrated in the section 4.2 in detail. The partition is made up of coalitions $\{C_i\}_{i=1}^M$ for $1 \leq M \leq N$;

(s2) Random sample the coalition $C \in \Pi$ from the multinomial distribution $\Gamma$ (in the section 4.2), and set $U \leftarrow C$ as the new global structure knowledge.

It iterates among steps (c1), (c2), (s1), and (s2) until reaching the final epoch. We can observe that the private node features of $\{S_i\}$ at the local client would not be uploaded to the server. On the server side, we can just obtain the public structure knowledge $\{U_i^*\}$.

Note that the bi-level optimization problem (6) has two targets. The local function $F_\alpha^i(S_i)$ aims to detect private anomalies using only the data of local data owner $i$. The alignment function $Q_\sigma^i(S_i^*, U_i^*)$ aims to federate these private anomalies by aligning the local private knowledge on the public global knowledge. We can observe that the two functions exhibit at two scales, and introduce a hyperparameter $\lambda$, $\max_{U \subseteq G_0} \sum_{i \in [N]} \max_{S_i \subseteq G_i} F_\alpha^i(S_i) + \lambda Q_\sigma^i(S_i, U)$, controls the interpolation between the two functions. The hyperparameter $\lambda$ is related to the specific form of $Q$, e.g., $Q$ as a regularization term, and the theoretical properties of FEDGRAPH is proved under the general setting without $\lambda$. When the value of $\lambda$ is set to 0, our algorithm performs as local knowledge detection tasks for each local data owner. Our algorithm focuses on more federated anomalies with increasing $\lambda$.

**Relation to Ditto federated learning** Li et al. (2021). We do not consider the network structure $\{G_i\}$, and each $G_i$ can be transformed to a vector $v_i$, e.g., $v_i(k) \leftarrow 1$ if the node $k$ is abnormal in $G_i$, and $v_i(k) \leftarrow 0$ otherwise. The vector $w$ is derived from $G_0$. We can take $Q$ as a regularization term, $\lambda/2 \cdot \| Av_i - w \|^2$. The optimization problem is to minimize $-F_i(v_i) + \lambda/2 \cdot \| Av_i - w^* \|^2$ for $v_i$ with each owner, subject to $w^* = \arg\min_w 1/N \cdot \sum_{i \in [N]} \| Av_i - w \|^2$. Our algorithm can be reduced to Ditto. Thus we propose a general algorithm framework for federated graph learning tasks.

### 4.4 EXTENSIVE DISCUSSION

We extensively discuss the federated graph learning problem (1) from *same* and *different* graph-structures between clients and server. In this new paradigm for the problem, we term the structure knowledge (i.e., subgraphs) as the "*parameters*" in the global models.

In the same graph-structure setting ($\forall i, \mathcal{G}_i = \mathcal{G}_0$), models can explicitly capture the topological structure and identify anomalies by measuring the reconstruction errors of nodes Ding et al. (2019). Formally, for an $L$-layer GNN, its $l + 1$-th layer can be formulated as

$$\mathbf{h}^{l+1}(v) = \text{UPD}^l\Big(\mathbf{h}^l(v), \text{AGG}^l\Big(\{\mathbf{h}^l(u) : \forall u \in \mathcal{N}(v)\}\Big)\Big)$$

where $\mathbf{h}^l(v)$ is the representation vector of node $v$ at the $l$-th layer, $\mathcal{N}(v)$ represents the set of node $v$'s neighbors. $\mathbf{h}^1(v)$ is initialized as $\mathbf{X}(v)$. The topological structure of $\mathcal{G}$ can be represented by an adjacency matrix $\mathbf{A}$, where $\mathbf{A}_{i,j} = 1$ if $(v_i, v_j) \in E$, otherwise $\mathbf{A}_{i,j} = 0$. We denote $\mathbf{H} = \{\mathbf{h}^3(v) : v \in V\}$ as all node latent representations. The problem (1) can be reformulated as:

$$\mathcal{L}(\mathcal{G}) = (1 - \alpha)\|\mathbf{A} - \sigma(\mathbf{H}\mathbf{H}^T)\|_F^2 + \alpha\|\mathbf{X} - f_{Relu}(\mathbf{H}, \mathbf{A})\|_F^2 \tag{5}$$

where $\alpha$ is a controlling parameter that balances the graph structure and node feature learning impacts. $\sigma$ is a sigmoid function that computes the probability of an edge $A_{u,v} = 1$ for $u, v \in V$, and its output is the estimated adjacency matrix. In the contrastive self-supervised learning framework, the objective

function can be reformulated as $\mathcal{L}(\mathcal{G}) = -\sum_v \mathbf{Y}(v) \log (F(\mathbf{X}(v))) + (1 - \mathbf{Y}(v)) \log (1 - F(\mathbf{X}(v)))$ where $\mathbf{Y}(v)$ is label of node $v$ and $F$ is the contrastive learning model Liu et al. (2022). After the federated learning epochs for the $i$-th client, we can derive the structure knowledge sequence from $\mathcal{G}_i, S^{(1)}, S^{(2)}, S^{(3)}, \cdots$ which are applied to the problem (5) and we achieve each objective function $\mathcal{L}^i(\mathcal{G}_i), \mathcal{L}^i(S^{(1)}), \mathcal{L}^i(S^{(2)}), \mathcal{L}^i(S^{(3)}), \cdots$ where the local model is addressed optimally at each epoch.

In the different graph-structure setting ($\exists i, \mathcal{G}_i \neq \mathcal{G}_0$), the structure knowledge provided by the server can be different from graph-structure in clients. We employ network alignment-based models in clients for the structure knowledge $U \subseteq \mathcal{G}_0$ can be aligned on the local knowledge $\forall i, S_i \subseteq \mathcal{G}_i$, where $S \subseteq \mathcal{G}$ denotes a subgraph $S$ of $\mathcal{G}$ whose vertex set $V_S$ and edge set $E_S$ are subset of $V, E$. In this setting, each client has its *exclusive model* to identify the subgraph $S_i$ from $(\mathcal{G}_i, \mathbf{X}_i)$, and the model is restrictive to share with outside. Here, we exemplify the problem (1) with the *federated anomaly subgraphs detection task* in detail.

We defines the empirical p-value corresponding to each node $v \in V$ (Wu et al., 2018; Chen & Neill, 2014), the smaller the p-value, the more abnormal the node. One global anomaly $\{S_1, S_2, \ldots, S_N\}$ (e.g., anomaly users in different social networks) is distributed as local anomalies on multiple clients. We denote $[N]$ as the set of $\{1, 2, \cdots, N\}$. The local anomaly $S_i$ for $i \in [N]$ can be identified by maximizing the local exclusive model $F_\alpha^i(S_i)$ with the significant level $\alpha$. The $\mathcal{G}_0$ in the server bridges the gap among isolated private data in clients, and the structure knowledge (i.e., public anomaly) $U \subseteq \mathcal{G}_0$ is associated with local private anomalies. A local alignment function $Q_\sigma^i(S_i, U)$ is employed to measure the similarity between $S_i$ and $U$ for $i \in [N]$. For anomaly subgraphs $S_i \subseteq \mathcal{G}_i$ and $U \subseteq \mathcal{G}_0$, each private data owner exclusively defines the alignment matrix $A^i \in \{0, 1\}^{n \times n}$, where $A_{uw}^i \leftarrow 1$ if the value of $(u, w)$ is greater than the alignment threshold $\sigma$ (i.e., a constant) with $u \in V_{S_i}$ and $w \in V_U$, $A_{uw}^i \leftarrow 0$ otherwise.

The federated anomaly detection task (1) across multiple clients can be reformulated as follows:

$$\min_{U \subseteq \mathcal{G}_0} \sum_{i=1}^N \mathcal{L}^i(\mathcal{G}_i; U) = \min_{U \subseteq \mathcal{G}_0} \sum_{i=1}^N -(\lambda Q_\sigma^i(S_i, U) + F_\alpha^i(S_i)) \tag{6}$$

where the private anomaly subgraph $S_i \subseteq \mathcal{G}_i$ is measured by $F$ with the significant level $\alpha$ (e.g., 0.15). The alignment score between $S_i$ and the public anomaly $U$ is measured by $Q$ (i.e., a distance function between $S_i$ and $U$) with the predefined alignment threshold $\lambda$ and $\sigma$ (e.g., 1.0, 0.8). However, GNN models can explicitly use the topological structure and nodal features seamlessly. Our proposed paradigm for federated graph learning can be applied to various tasks for clients who have already employed the models. The clients do not need to change the production-in-use models for the federated graph learning setting.

## 5 EXPERIMENTS

### 5.1 EXPERIMENTAL SETUP

**Datasets.** We evaluate the performance of the proposed framework on three widely used benchmark datasets and two real-world datasets tailored for anomaly detection. The datasets include citation network datasets (Cora, CiteSeer, ACM), computer networks and multi-road traffic datasets. The statistics of these datasets are illustrated in Table 4 with detailed descriptions. Since the base graphs lack ground-truth anomalies, we inject structural and attribute anomalies for each dataset.

**Our Model and Baselines.**

In our experiment, we examine three variants of our framework: **FedGraph(Dom)** and **FedGraph(CoLA)**, which incorporate the DominantDing et al. (2019) and CoLALiu et al. (2022) anomaly detection algorithms, respectively, and are aimed at non-deep learning approaches for anomaly subgraph detection. Our baselines include **FedAvg**McMahan et al. (2017), a distributed machine learning optimization algorithm; **Fed-Pub**Baek et al. (2023), which enhances local GNNs through functional embeddings; **Loc-Dominant**, a deep graph autoencoder for detecting anomalies in attributed networks; **Loc-CoLA**, a GNN-based contrastive learning approach for node anomaly scoring; and three methods—**ASD-FT**Sun et al. (2020), **TSPSD**Wu et al. (2016), and **NPHGS**Chen & Neill (2014)—that utilize network alignment for detecting anomalies across different graph structures. The final assessment accuracy is determined by averaging the accuracy values across these methods.

Table 1: Accuracy evaluation of our federated graph learning algorithms and baselines on Cora, CiteSeer, and ACM datasets, the mean degree of these datasets are 27.30, 19.80, 65.50. All results are based on the top-$k$ nodes identified as anomalies, with $k = 10$. ("-" denotes that no anomaly subgraph is detected.)

| # Datasets | Cora | | | | CiteSeer | | | | ACM | | | |
|---|---|---|---|---|---|---|---|---|---|---|---|---|
| # clients | 3 | 6 | 10 | 20 | 3 | 6 | 10 | 20 | 3 | 6 | 10 | 20 |
| Loc-Dominant | 23.14 | 29.32 | 9.82 | 11.94 | 11.11 | 12.50 | 13.31 | 9.51 | - | 35.72 | 27.69 | 11.24 |
| Loc-CoLA | 64.48 | 27.13 | 14.15 | 11.94 | - | 16.24 | 15.60 | 9.51 | 27.73 | 15.22 | 8.96 | 13.68 |
| FedAvg | 84.61 | 71.03 | 81.23 | 54.61 | 81.82 | 81.09 | 58.56 | 73.75 | 70.00 | 35.71 | 25.83 | 25.99 |
| Fed-PUB | 85.38 | 70.70 | 68.46 | 61.10 | 86.36 | 81.98 | 66.30 | 74.55 | 71.43 | 45.00 | 37.22 | 45.88 |
| **FedGraph(Dom)** | 16.67 | 63.63 | **90.48** | 69.23 | - | 14.29 | 38.46 | 44.11 | 33.33 | 43.24 | 41.67 | 35.48 |
| **FedGraph(CoLA)** | **85.71** | **93.48** | 85.45 | **91.57** | **90.48** | **88.00** | **70.73** | **76.67** | **75.00** | **61.76** | **58.70** | **68.63** |

Table 2: Evaluating our different graph-structure method (FedGraph) and baselines on the Computer Network dataset, with the number of clients set to 3, 6 and 10.

| Method | Average Precision | Accuracy | | | |
|---|---|---|---|---|---|
| | | 3 | 6 | 10 | Average Accuracy |
| ASD-FT | 94.00 | 89.12 | 81.89 | 76.02 | 80.07 |
| TSPSD | 95.00 | 90.46 | 79.70 | 75.54 | 78.74 |
| NPHGS | 97.00 | 83.23 | 76.24 | 73.67 | 74.48 |
| **FedGraph** | **99.00** | **95.63** | **88.26** | **82.15** | **86.22** |

## 5.2 EXPERIMENTS RESULTS

The feasibility of our federated graph learning framework allows us to categorize the accuracy assessment in our experiment into two sections: same graph-structure comparison (Table 1) and different graph-structure comparison (Table 2). The same graph-structure comparison is conducted on the Cora, CiteSeer, and ACM datasets, while the different graph-structure comparison is performed on the computer network dataset. It's important to note that the corresponding baselines differ across two types of comparisons.

### 5.2.1 EVALUATION ON SAME GRAPH-STRUCTURE METHODS

In this evaluation experiment, we use FedGraph(Dom) and FedGraph(CoLA) to compare against baselines. The accuracy metrics, shown in Table 1, are calculated with anomalies ranked in the top-k (k = 10). We divide datasets into 3, 6, 10, and 20 parts to create scenarios with varying client numbers. Our approaches consistently achieve the highest accuracy scores across all clients and datasets. Notably, in the Cora dataset with 6 clients, FedGraph(CoLA) reaches an accuracy of 0.9348, outperforming the competitive baseline Fed-PUB by at least 22.45%. In the CiteSeer dataset, our approach exceeds competitive baselines by [2.12%, 10.45%]. In ACM, FedGraph(CoLA) consistently achieves the highest accuracy.

On the other hand, Loc-Dominant and Loc-CoLA obtained lower accuracy in Cora and CiteSeer, possibly due to the lower mean degrees of the two datasets. After dividing the dataset into several parts, some neighbor relationships were lost, and this loss of topology information significantly impacted the training and detection results of Dominant and CoLA when run locally. ACM exhibits a closer neighbor relationship among its nodes, which explains why Loc-Dominant and Loc-CoLA achieve higher accuracy scores.

### 5.2.2 EVALUATION ON DIFFERENT GRAPH-STRUCTURE METHODS

To validate the effectiveness of FedGraph on various graph-structure networks, we split the Computer Network dataset into as many subsets as clients. Besides FedGraph, all baseline methods are non-deep learning anomaly detection algorithms. As shown in Table 2, our approach outperforms the others, achieving the highest precision score of 99%, detecting nearly all anomalies, and surpassing the best baseline by 2%. Additionally, our accuracy exceeds the competitive baseline by [5.1%, 6.37%], with an average accuracy (AvgACC) higher than the baselines by at least 6.15%.

Table 3: Ablation Study for Label-Trick with FedGraph(Dom)

| Dataset | Label-Trick | | non Label-Trick | |
|---|---|---|---|---|
| | # Subgraph size | Accuracy | # Subgraph size | Accuracy |
| Cora | 21 | 90.48 | 16 | 68.75 |
| CiteSeer | 10 | 38.46 | 12 | 33.33 |
| ACM | 6 | 33.33 | 7 | 28.57 |

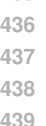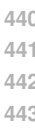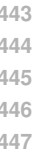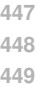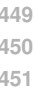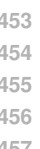

Figure 3: Robustness of our approaches. The noise level is set to the range of $[5\%, 30\%]$. (a) and (b) illustrate the algorithm's robustness in terms of accuracy, while (c) and (d) depict the change in the size of the detected anomaly subgraph.

**Summary:** In experiments involving same graph-structure methods, for a dataset that has no obvious anomaly, e.g., Cora, CiteSeer, or ACM, algorithms can learn the anomaly features through the topology structure and nodes' attributes, resulting in detection outcomes that exhibit some degree of randomness. However, in the context of the Computer Network dataset, the anomaly feature can be represented by the P-value, which is calculated based on the volume of traffic between nodes. Therefore, the different graph-structure methods detect the anomaly subgraph with the P-value and present higher performance scores than the same graph-structure methods. Table 1 and Table 2 illustrate these differences. Table 1 and Table 2 illustrate these differences.

## 5.3 ABLATION STUDY

Label-Trick is a key technique in our approach. In this ablation experiment, we use FedGraph(Dom) to demonstrate the importance of Label-Trick. We evaluate its performance on the Cora, Citeseer, and ACM datasets by adding and removing the Label-Trick component. As shown in Table 3, FedGraph(Dom) achieves higher accuracy scores with Label-Trick compared to without it, validating its effectiveness in detecting anomalous subgraphs iteratively on both client and server sides. The contribution of Label-Trick to the size and accuracy of the detected anomalous subgraphs is significant.

## 5.4 ROBUSTNESS STUDY

We validate the robustness of the FedGraph framework by introducing noise nodes into the Cora and Citeseer datasets at proportions of 5%, 10%, 20%, and 30%. These noise nodes are injected with structural and attribute anomalies while retaining their original labels. We observe the final sizes of anomalous subgraphs and accuracy scores. As shown in Figure 3, although the accuracy of both FedGraph(Dom) and FedGraph(CoLA) decreases with more noise nodes, it remains acceptable. Notably, at a 10% noise proportion, both methods show a significant decrease, but as the proportion increases further, they stabilize, indicating satisfactory robustness in the FedGraph framework.

## 6 CONCLUSION

We introduced a novel paradigm FedGraph, which addresses the model isolation to federated graph task challenge. The proposed FedGraph algorithm framework considered the graph structure as the bridge connecting private node features. It leveraged the graph structure as structural knowledge, linking all task-related knowledge in clients. Our extensive experiments demonstrated that FedGraph significantly outperformed other state-of-the-art federated learning algorithms in the context of anomaly detection tasks.

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

# A    APPENDIX

## A.1    FEDERATED LEARNING LIMITATIONS

Traditional federated learning cannot be applied in scenarios where both data and algorithms are private. Specifically, let $\mathcal{D}_i$ represent the private data of client $i$, and $f_i$ represent the private algorithm of client $i$. Since $f_i$ is private, the model parameters $\theta_i$ learned by each client's algorithm are inconsistent across different clients. Traditional federated learning McMahan et al. (2017) relies on the assumption that model parameters $\theta_i$ can be shared and aggregated across clients to create a global model $\Theta = \sum_{k=1}^{K} \frac{|D_k|}{\sum_{i=1}^{K} |D_i|} \theta_i$, where $K$ is the number of clients. However, due to the inconsistency in $\theta_i$ arising from private $f_i$, this aggregation is not feasible, rendering traditional federated learning ineffective. The new federated graph learning paradigm proposed in this paper addresses this issue by focusing on the structure of graph data. It utilizes graph structures to connect knowledge from different clients, thereby overcoming the isolation caused by private data $\mathcal{D}_i$ and private algorithms $f_i$. This approach bridges the knowledge gap without relying on the sharing of parameters $\theta_i$.

## A.2    DATASETS

**1) Cora**. Cora dataset is an academic citation network dataset, which plays a key role in graph federation learning research, where nodes represent academic papers and edges indicate citation relationships between papers. The features of Cora dataset are bag-of-words modeling word frequency statistics for each paper. Cora has been widely used for evaluating the performance of federation learning algorithms on the task of anomalous subgraph detection. **2) CiteSeer**. The CiteSeer dataset is a scholarly citation network dataset, and its network structure and attribute features are similar to the Cora dataset. **3) ACM**. The ACM dataset is also a scholarly citation network dataset, and its network structure and attribute characteristics are similar to the above two datasets. **4) Computer Network**. The computer network dataset represents a computer traffic networks, where nodes correspond to websites and edges indicate the presence of an attack relationship between two websites. From May 31, 2014, to May 31, 2015, the computer network was divided into six private networks $(\mathcal{G}_1, \cdots, \mathcal{G}_6)$ at two-month intervals, each demonstrating distinct anomalous properties. Subsequently, we conducted correlated anomaly detection on these attributed networks. **5) Multi-Road Traffic**. The multi-road traffic dataset comprises the Car-hailing itinerary network, Bike-sharing itinerary network, and Subway network crawled on December 21, 2019. In the Car-hailing itinerary network and Bike-sharing itinerary network, nodes represent the starting and ending points of an itinerary, while edges connect the nodes of the respective itineraries. In the Subway network, nodes represent stations, and edges depict subway lines connecting these stations.

Table 4: The summary of the dataset statistics.

| Dataset | #Nodes | #Edges | #Attributes | #Anomalies |
|---|---|---|---|---|
| **Cora** | $2,708$ | $5,429$ | $1,433$ | $300$ |
| **Citeseer** | $3,327$ | $4,723$ | $3,703$ | $300$ |
| **ACM** | $3,025$ | $13,128$ | $1,870$ | $300$ |
| **Computer Network** | $134,225$ | $181,771$ | $367$ | $18,673$ |
| **Multi-road Traffic** | $75,963$ | $388,643$ | $24$ | - |

### A.2.1    ANOMALY INJECTION

Since the base graphs lack ground-truth anomalies, we employ the following methods to inject structural and attribute anomalies for each dataset.

**1) Injecting structural anomalies:** A method involving perturbation of network topology is utilized to inject the structural anomalies. Specifically, we generate $q$ small anomalous clusters ($q \in \mathbb{R}$). Initially, $q$ nodes are randomly selected from the node-set $V$, and random walk approach is utilized to sample $q$ connected subgraphs. Subsequently, these $q$ connected subgraphs are transformed into $q$ fully connected subgraphs. Suppose each fully connected subgraph includes $p$ nodes; these selected $p$ nodes are labeled as "Structural Anomaly Nodes." In total, $p \times q$ structural anomalies are injected. The quantity of injected anomalies is adjusted based on the dataset size.

**2) Injecting attribute anomalies:** We perturb the attributes of nodes. To ensure an equal number of attribute and structural anomalies, we conduct $q$ random walks, each time selecting $p$ connected nodes as targets. Subsequently, for each selected node $v_i$, we randomly sample $k$ other nodes $V_c = (v_{c1}, ..., v_{ck})$ to form a candidate set. For each $v_c \in V_c$, we calculate the Euclidean distance between its attribute vector $x_c$ and the attribute vector $x_i$ of $v_i$. Following that, we select the node $v_{cj} \in V_c$ with the maximum Euclidean distance and replace the attribute $x_i$ with $x_{cj}$. In our paper, $p$ is set to 60, and $q$ is set to 3 for the ACM, Cora, and Citeseer datasets. One of the selected subgraphs is injected with anomalous structures.

## A.3    BASELINES

**1) FedAvg**McMahan et al. (2017) is an optimization algorithm for distributed machine learning. FedAvg preserves privacy, minimizes communication, and fosters collaboration. It locally trains models, uploads parameters to a central server, and performs weighted averaging to update the global model iteratively.**2) Fed-Pub**Baek et al. (2023) enhances local GNNs in a privacy-centric global graph by using functional embeddings and server-side aggregation. Personalized sparse masks, learned at each client, update relevant subgraphs, addressing missing links and privacy constraints without a single global model.**3) Loc-Dominant** is adjusted from DOMINANTDing et al. (2019) which employs a deep graph autoencoder method, leveraging both graph structure and features to detect anomalies in an attributed network, with separation by specified clients. The final assessment accuracy is obtained by calculating the average values of all accuracies.**4) Loc-CoLA** is adapted from CoLALiu et al. (2022), an anomaly detection algorithm emphasizes nodes and employs a Graph Neural Network (GNN)-based contrastive learning approach at the node-subgraph level. It calculates anomaly scores for nodes by assessing representations generated from nodes and subgraphs within positive and negative instance pairs. The detailed assessing method is similar to that of *loc-Dominant*.**5) ASD-FT**Sun et al. (2020). ASD-FT addresses sparse anomalous features in multilayer graphs by transferring information between layers. It detects anomaly subgraphs in one layer by analyzing anomalous features from another.**6) TSPSD**Wu et al. (2016). TSPSD reformulates non-parametric graph scan as Budget Price-Collecting Steiner Tree sub-problems. Efficient algorithms target anomalous subgraphs in fixed tree topology graphs, validated in diverse applications.**7) NPHGS**Chen & Neill (2014). NPHGS optimally detects social media events by leveraging the entire heterogeneous network structure, using a nonparametric scan statistic on a "sensor" network.

## A.4    CASE STUDY

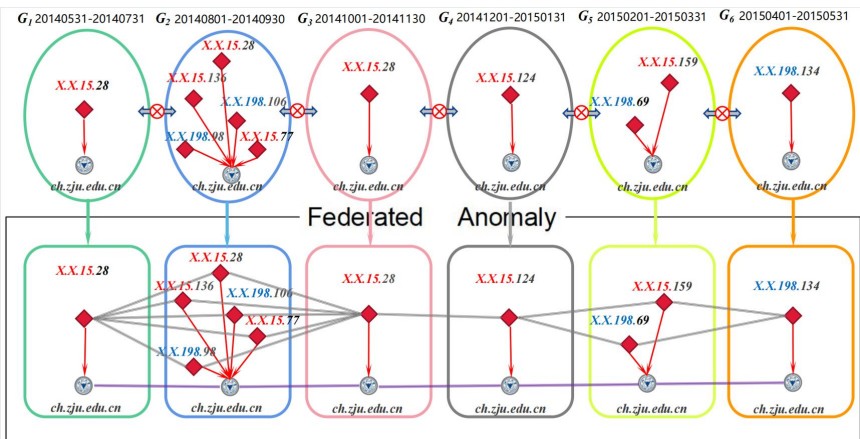

Figure 4: **A set of related anomalous IPs was detected by our method**. The detected IPs have potential correlations (e.g., at the same place). The site *www.ch.zju.edu.cn* was attacked mainly from two network segments *x.x.15.(28-159)* and *x.x.198.(69-134)*. The addresses of these IPs were all at the same place (i.e., *Shanxi, China*). The attack methods were all *Nginx Attack*.

### A.4.1    CASE STUDY IN COMPUTER NETWORK DATASET

Run FEDGRAPH on the computer network dataset, and set $\alpha = 0.15$, $\sigma = 0.8$.

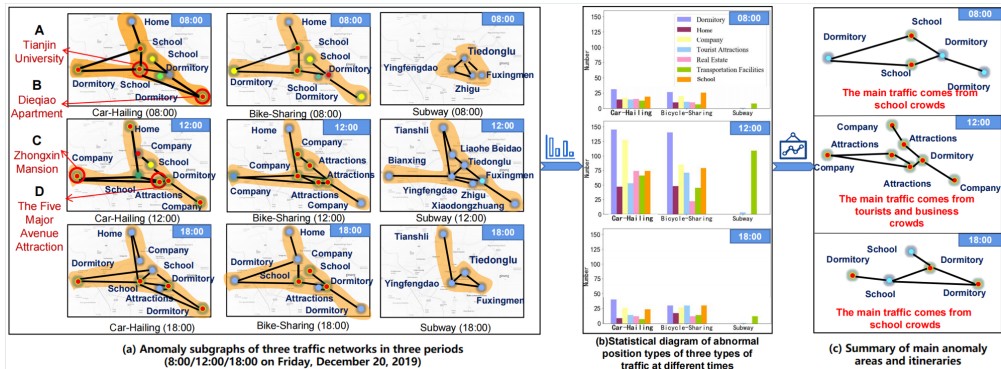

Figure 5: Correlated anomalies at 8:00, 12:00, and 18:00 on December 21, 2019. (a) depicts a heat map illustrating the distribution of anomalous traffic networks. Each hot spot represents an anomalous area, and the redder the color, the higher the anomaly level of the area. The black lines indicate that there are itineraries between the two areas, and the dark blue text indicates the main position type of the start and end points of the itineraries included in the area. (c) is a summary of the anomalies at three periods in (a). From analyzing the anomaly areas and itineraries, it's evident that the morning and evening traffic primarily comprises school crowds, whereas at noon, it predominantly consists of business crowds and tourists.

**1) Discovery of related anomalous IP group**: Our algorithm can identify the anomalous IP group and uncover the hidden attacking IP information (Figure 4). FEDGRAPH can discover potential anomalous anchor links among networks. By summarizing the anchor nodes associated with these anchor links, we can pinpoint an anomalous IP group. Despite these IPs appearing at different times, their attack patterns remain consistent. Upon analyzing their log data, we observed that these IPs originate from several distinct network segments. The similarity in attack methods and target locations suggests that these IPs may originate from the same source of attack. Our approach effectively identifies cyber threats and enhances cyber attack prevention measures.

**2) Prediction of cyber attacks**: We devided the computer network dataset into six networks according to temporal attributes, labeled as $\mathcal{G}_i$ $(i = 1, 2, ..., 6)$. We designate $\mathcal{G}_6$ as an attributeless network by assigning p-value of 1 to all its nodes. We then combine it with the other networks as input for FEDGRAPH to generate its anomaly subgraph, $S_6$. Regard $S_6$ as the prediction result, which summarizes the IPs that may attack the website during the period of $\mathcal{G}_6$. We compare it with the real attacks that occurred, and our algorithm can detect the anomalous situation of the target network through networks with sufficient anomalous characteristics, even if the target network does not have anomalous information.

A.4.2 CASE STUDY IN TRAFFIC DATASETS

We perform FEDGRAPH on the multi-road traffic dataset (i.e., car-hailing, bike-sharing, and subway networks). The location information of anomaly detection results is acquired from the Points of Interest (POI) data. Thus, we visualize this information on a geographic map. We selected three networks corresponding to the same period (8:00/12:00/18:00) and set $\alpha = 0.05$ and $\sigma = 0.8$.

**Discovery of real events**:

The anomaly detection results, denoted as the "mapping graph" below, reveal distinctive patterns. In the Bike-Sharing network mapping graph, anomalies tend to concentrate around schools, hospitals, and parks, aligning with the city's morning activity centers. This distribution authentically reflects real-world scenarios. Analyzing the Car-Hailing and Bike-Sharing datasets' mapping graph reveals a heightened concentration of anomalies near Xigu Park compared to other parks. Further investigation into the Weibo dataset uncovered posts predominantly detailing morning visits to Xigu Park on that particular day. In-depth analysis of detection results from both datasets provides additional insights. Notably, the mapping graph of Car-Hailing, Bike-Sharing, and Subway datasets indicates clusters of anomalies around Tianjin University, Nankai University, and Tianjin No. 21 Middle School. Interestingly, anomalies are conspicuously absent near Tianjin No. 1 Middle School and Tianjin No. 2 Middle School. After analyzing relevant data, we concluded: The dataset, dated December 21, 2019, coincides with the National Unified Entrance Examination for Master's graduates. Anomalies

clustered near schools are linked to the exam. Candidates from Tianjin University and Nankai University primarily walked or rode shared bikes. Those from Tianjin No. 21 Middle School, mostly off-campus, used Car-Hailing services. Anomalies are abundant near these sites due to distinct travel patterns. Schools like Tianjin No. 1 and No. 2 show no anomalies. This analysis highlights the importance of multiple datasets in yielding interpretable results, aligning with FedGraph's goal of detecting more anomalies.

