# OpenReview forum: "FedGraph: A New Paradigm for Federated Graph Learning"
_ICLR.cc/2025/Conference — Submitted to ICLR 2025_

### Official Review · Reviewer_HE9m · 2024-10-29

**Soundness:** 2
**Presentation:** 2
**Contribution:** 2
**Rating:** 3
**Confidence:** 2

**Summary:**

The paper introduces FEDGRAPH, a novel federated graph learning paradigm designed to tackle the challenges associated with model isolation in federated graph tasks. By sharing graph structures, FEDGRAPH effectively bridges task-specific knowledge across clients. Extensive experimental results demonstrate that it significantly outperforms existing state-of-the-art federated learning algorithms in anomaly detection tasks.

**Strengths:**

1. This study is the first to investigate the problem of federated graph learning without the need for model parameter aggregation.
2. The experimental results show significant improvements.
3. The organization of the paper is clear and well-structured.

**Weaknesses:**

1. The definitions in the paper are somewhat confusing. For example, certain symbols are introduced without timely explanations, such as $\alpha$ in line 86, $\sigma$ in line 88, and $\mathcal{L}_i$ in line 150. Additionally, there appears to be a conflict between the statement in line 79, "We take the global graph structure as structural knowledge," and the definition provided in line 76, where 'knowledge' refers to subgraphs mined by algorithms. Furthermore, the authors do not clarify the difference between $S$ and $\textbf{S}$ in Equation 1.
2. In the related work section, the discussion of Federated Graph Learning lacks a comprehensive summary and synthesis, and recent literature from the past two years is notably absent.
3. The paper does not address the potential privacy and security concerns associated with structure sharing compared to parameter sharing, which could be significant.
4. There is a lack of pseudocode or experimental code provided to ensure the reproducibility of the experiments.
5. The analysis of hyperparameters is missing, which would be beneficial for understanding the experimental results.

**Questions:**

1. Which is more critical to protect: the model parameters or the structural features of the data?
2. Why not test it on more general graph tasks? How can it be extended to other graph tasks? Such as link prediction and graph classification.
3. How do the server and clients interact? Is there a need to transmit embeddings or other information from one client to another?
4. The primary downstream task discussed in this paper is graph anomaly detection, which is closely related to graph structure. Considering that other baseline graph structures are not visible, does this comparison lack fairness?
5. The equations in lines 313-314 are not numbered, and there is no explanation provided for UPD. What forms could UPD take?
6. Which part of Equation 5 corresponds to $\overline{S}$ in Equation 1, and why can it be transformed in that way?

---

### Official Review · Reviewer_RjLd · 2024-11-04

**Soundness:** 1
**Presentation:** 2
**Contribution:** 1
**Rating:** 3
**Confidence:** 3

**Summary:**

This paper studies the problem of model isolation to federated graph task, where graph models are different across clients. The authors propose FedGraph, a new paradigm for federated graph learning. It utilizes graph structure without private node features as structure knowledge to bridge all task-specific knowledge in clients. Extensive experiments validate the effectiveness of the proposed method on anomaly detection tasks.

**Strengths:**

1. This study employs the graph structure to bridge different models from clients.
2. Many graph learning models can be transferred to the proposed framework.
3. The proposed framework is effective and robust.

**Weaknesses:**

1. Sharing graph structure may breach privacy protection in FGL. In many cases, graph structure information is a very important part of graph data. Sharing graph structure can be viewed as critical privacy leakage. Therefore, the proposed problem with sharing graph structure may not make great sense in the real world.
2. The main problem is unclear. The authors should reconsider whether this study is for general federated graph learning or federated graph learning in anomaly detection. Section 4 and Section 5 mainly focus on anomaly detection, but the motivation of this study seems irrelevant to anomaly detection.
3. Important experiment details are missing. For example, the authors should specify how the graph datasets are split into multiple clients. In addition, the authors should specify what graph models are used in the experiment.
4. The phrase *Knowledge Graph* is a specialized term in graph learning and represents a type of heterogeneous graph. The authors may replace it to avoid ambiguity.

**Questions:**

1. Can the proposed method be applied to tasks other than anomaly detection?
2. What is Label-Trick in Section 5.3? The authors claim that Label-Trick is a key technique in the proposed approach, but it is not mentioned in the Methodology.

---

### Official Review · Reviewer_nMkQ · 2024-11-07

**Soundness:** 2
**Presentation:** 2
**Contribution:** 2
**Rating:** 3
**Confidence:** 3

**Summary:**

This paper presents a new challenge in federated learning on graphs, specifically focused on anomaly detection, by introducing the idea of “model isolation to federated graph tasks". In this setting, local graph structures are heterogeneous, and local algorithms (models) may vary. The authors propose a method that leverages structural knowledge to bridge task-specific knowledge. This is achieved by learning relevant public structures through the aggregation of local structures via network alignment, which is then shared back with the clients.

**Strengths:**

Different from most federated learning (FL) papers that share either models or embeddings, this paper solely shares graph structures. It aims to use these structures to bridge knowledge gaps across clients by separating structure learning from feature learning, thus protecting the privacy of local features and algorithms.

**Weaknesses:**

1. The scenarios are not clearly explained and could be elaborated further, especially regarding their relevance to this paper’s problem setting. Additionally, it is hard to tell what the key challenges are from the introduction.
2. The overall presentation needs improvement, as many sections require effort to understand. For example, the key points of each paragraph in the introduction are unclear, and the transitions lack logic and smoothness.
3. While the authors introduce the new concept of “model isolation to federated graph tasks,” it’s unclear how this differs from data/model heterogeneity in FL. Moreover, it’s essential to clarify how model isolation relates to structural knowledge.
4. Although this work is presented as a generic federated graph framework, it focuses on a single downstream task (anomaly detection) and experiments on limited datasets and methods. Narrowing the scope or making the key contributions clearer is necessary.
5. There exist privacy concerns regarding alignment, i.e., node identities stored on the server. Are these nodes anonymous? If so, how are they aligned with local node features? Additionally, privacy concerns exist for identities transmitted between the server and clients.
6. In different graph structure settings, the proposed FL paradigm is unrelated to local features and model training. Thus, claiming it addresses the challenges (characteristics) is not convincing, as it bypasses them. Also, there are no experiments involving different local algorithms.
7. There is no hyperparameter study regarding the important predefined alignment threshold, $\sigma$.
8. The paper lacks discussion on efficiency and privacy aspects.
9. There missing more recent related works.

**Questions:**

Q1. How does “Characteristic 2” (Parameter Mismatch) differ from model heterogeneity in FL?
Q2. Features are critical in anomaly detection. Can you explain why sharing only structures is sufficient?
Q3. What is the anchor link between S and U? The definition seems to be missing.
Q4. How does coalition partition relate to the set of public knowledge {U}? It’s unclear how coalition partition is derived from the {U}. From the authors' sentence “computing graph connected components in the global graph G0”, the partition of $\Pi$ can differ from the partition of {U}, then how can the contribution be calculated?
Q5. Why is federated learning needed in a scenario where graph structures, local algorithms, and tasks differ significantly?
Q6. How is the received relevant public knowledge used locally while the local clients still train models using the original local features and structures?
Q7. If the focus is on graph anomaly detection, using the same algorithms seems feasible. What is the motivation behind not sharing models in FL and using structures to bridge the “model gap”?
Q8. Why not use common anomaly detection benchmark datasets?
Q9. In Table 1, why does accuracy for the same setting vary so widely across methods (e.g., from 9.82 to 90.48)?

---

### Official Review · Reviewer_n9iz · 2024-11-08

**Soundness:** 3
**Presentation:** 1
**Contribution:** 2
**Rating:** 3
**Confidence:** 4

**Summary:**

This paper introduces FedGraph, a novel paradigm for federated graph learning that aims to address both data heterogeneity and model isolation challenges in federated graph learning. Specifically, the authors propose to use graph structures as a bridge to connect private features and algorithms across different clients, transforming private knowledge into public structural knowledge for sharing, and design mechanisms for local structure knowledge updates and adaptive sampling.

The main contributions can be summarized as follows:

1. The paper is the first to propose a federated graph learning paradigm that does not require parameter aggregation.
2. It supports the migration of various existing graph learning models to the FedGraph framework.
3. The experimental results on five datasets demonstrate significant improvements compared to previous methods.

**Strengths:**

1. The research motivation is well-justified and addresses a genuine problem that requires solution in real-world applications.

2. The experimental results are impressive, demonstrating significant improvements over existing state-of-the-art methods.

**Weaknesses:**

1. The problem formalization section lacks a clear description of the specific task types addressed on graph data, and instead focuses primarily on formalizing the challenges faced. The purpose of this section is unclear as it fails to convey the author's task objectives, making it inconsistent with the section's intended purpose.

2. The main illustrative figures only show formal definitions of federated graph learning problems without demonstrating the actual algorithmic process or providing visual representations. The figure descriptions lack any concrete introduction to the algorithm content. No effective information about the proposed FedGraph algorithm can be obtained from these figures.

3. The description of the main algorithmic components is unclear and lacks coherence. Each sub-module describes its own functionality without explaining the connections between modules. The paper also lacks a complete and detailed description of the federated learning process, particularly regarding the interactions between clients and server and their respective behaviors in each round.

4. The methodology section appears to implement each module using existing algorithms from previous work. The paper lacks a clear description of its own integrative contributions and innovations.

5. The experimental evaluation is limited to small-scale datasets, lacking validation on larger-scale datasets and more clients setting. Furthermore, the experimental objectives remain inadequately described, making it difficult to fully assess the method's effectiveness and generalizability.  it was not until the appendix that we learned that the experimental part of the article is an anomaly detection task.

**Questions:**

1. Please provide a formal definition of the specific tasks addressed in federated graph learning. The current formulation lacks clarity on what types of graph-based tasks the proposed framework aims to solve.

2. In the FedGraph algorithm, what are the specific operations performed by clients and server in each round of federated learning? Please provide a clear description of the sequence of operations and how interactions occur between clients and server. The current description lacks a systematic presentation of this workflow.

3. The paper claims innovation in parameter-free aggregation, but how is this specifically achieved through different modules? How does this approach differ from the previous methods cited in the methodology section?

---

### Meta-Review · Area_Chair_4g91 · 2024-12-20

**Metareview:**

The paper introduces FedGraph, a federated graph learning paradigm that bridges task-specific knowledge across clients by sharing graph structures rather than model parameters. This approach addresses the challenges of data heterogeneity and model isolation, particularly for anomaly detection tasks. The experimental results show that FedGraph outperforms existing methods, demonstrating its effectiveness in federated settings.

While the proposed FedGraph model shows promising results, the reviewers have identified several weaknesses that need to be addressed:

1. The problem formalization is unclear and lacks focus on the specific types of graph tasks the framework addresses, making it hard to understand the primary objective.
2. The illustrations and figures are not informative and fail to convey the actual algorithmic process, limiting their usefulness in understanding FedGraph.
3. The description of the algorithmic components lacks clarity, with insufficient explanation of how the different modules connect or interact during the federated learning process.

Based on these weaknesses, we recommend rejecting this paper. We hope this feedback helps the authors improve their paper.

**Additional Comments On Reviewer Discussion:**

The authors did not provide a rebuttal.

---

### Decision · Program_Chairs · 2025-01-22

Reject